# Genomic Characteristics and Phylogenetic Analyses of a Multiple Drug-Resistant *Klebsiella pneumoniae* Harboring Plasmid-Mediated *MCR-1* Isolated from Tai’an City, China

**DOI:** 10.3390/pathogens12020221

**Published:** 2023-01-31

**Authors:** Qinqin Liu, Zhiyun Guo, Gang Zhu, Ning Li, Guanchen Bai, Meijie Jiang

**Affiliations:** 1Department of Hematology, The Affiliated Tai’an City Central Hospital of Qingdao University, Tai’an 271000, China; 2College of Animal Science and Veterinary Medicine, Shandong Agricultural University, Tai’an 271000, China; 3Department of Emergency, The Second Affiliated Hospital of Shandong First Medical University, Tai’an 271000, China; 4Department of Laboratory Medicine, The Affiliated Tai’an City Central Hospital of Qingdao University, Tai’an 271000, China

**Keywords:** *K. pneumoniae*, *MCR-1*, plasmid, whole-genome sequencing, functional annotation

## Abstract

*Klebsiella pneumoniae* is a clinically common opportunistic pathogen that causes pneumonia and upper respiratory tract infection in humans as well as community-and hospital-acquired infections, posing significant threats to public health. Moreover, the insertion of a plasmid carrying the mobile colistin resistance (MCR) genes brings obstacles to the clinical treatment of *K. pneumoniae* infection. In this study, a strain of colistin-resistant *K. pneumoniae* (CRKP) was isolated from sputum samples of a patient who was admitted to a tertiary hospital in Tai’an city, China, and tested for drug sensitivity. The results showed that KPTA-2108 was multidrug-resistant (MDR), being resistant to 21 of 26 selected antibiotics, such as cefazolin, amikacin, tigecycline and colistin but sensitive to carbapenems via antibiotic resistance assays. The chromosome and plasmid sequences of the isolated strain KPTA-2108 were obtained using whole-genome sequencing technology and then were analyzed deeply using bioinformatics methods. The whole-genome sequencing analysis showed that the length of KPTA-2108 was 5,306,347 bp and carried four plasmids, pMJ4-1, pMJ4-2, pMJ4-3, and pMJ4-4-MCR. The plasmid pMJ4-4-MCR contained 30,124 bp and was found to be an IncX4 type. It was the smallest plasmid in the KPTA-2108 strain and carried only one resistance gene *MCR-1*. Successful conjugation tests demonstrated that pMJ4-4-MCR carrying *MCR-1* could be horizontally transmitted through conjugation between bacteria. In conclusion, the acquisition and genome-wide characterization of a clinical MDR strain of CRKP may provide a scientific basis for the treatment of *K. pneumoniae* infection and epidemiological data for the surveillance of CRKP.

## 1. Introduction

*Klebsiella pneumoniae* is a clinically common Gram-negative conditional pathogen that can infect humans and a wide range of domestic animals [1]. In healthy populations, *K. pneumoniae* colonizes mainly the gastrointestinal tract, with a small concentration in the oropharynx [2]. In immunocompromised or immunodeficient patients, *K. pneumoniae* can invade the blood or tissues and cause a variety of infections, including urinary tract infections, bacteremia, pneumonia, and liver abscesses [3]. In addition, *K. pneumoniae* is a major pathogen in ventilator-associated pneumonia (VAP) and intensive care unit (ICU)-acquired pneumonia [4]. As a result, *K. pneumoniae* has received widespread attention.

Colistin is considered to be the last line of defense for the treatment of clinically Gram-negative bacterial infections, but the prevalence of colistin resistance has become an important problem worldwide [5]. In 2015, the plasmid-mediated colistin-resistance gene (MCR)-1 was reported for the first time internationally, and plasmids carrying the *MCR-1* gene were found to be transferable by inter-bacterial conjugation [6]. However, *MCR-1*-carrying plasmids have also been reported in many bacteria, such as *Salmonella* and *Acinetobacter baumannii* [7,8]. Recently, one study reported that plasmids harboring *MCR-1* could be horizontally transmitted between intestinal flora, allowing recipient bacteria to acquire colistin resistance ability [9]. In hospitals, pathogenic bacteria can acquire and spread resistance genes among themselves through bacterial conjugation, even in the absence of antibiotics. This phenomenon also occurs in the community, leading to the rapid emergence of multidrug-resistant (MDR) bacteria and the increased infection of susceptible populations [10].

*K. pneumoniae* carrying IncX4-type plasmids of *MCR-1* are present in different hosts (human, pig, poultry) and environments in many different countries, including China [11,12,13]. *K. pneumoniae* is a pathogen with a high detection rate of hospital-acquired infections. To date, clinical strains of *K. pneumoniae* carrying *MCR-1* have been isolated from patients in different countries such as Algeria, Belgium, Brazil, Colombia, France, Germany, India, Italy, Laos, South Africa and Vietnam, indicating that *K. pneumoniae* carrying *MCR-1* is spreading in healthcare facilities worldwide [14]. Most of the clinical reports on colistin-resistant pneumococci (CRKP) have focused on the analysis of isolation rates and resistance phenotypes of CRKP. The few reports that have studied *K. pneumoniae* carrying the *MCR-1* plasmid have not sufficiently annotated the genomic features of this bacterium and the function of some important genes. Given the impediments to clinical treatment and the negative impact on public health, knowledge of the genome-wide characteristics of *K. pneumoniae* can help to assess its risk more comprehensively [15]. 

Whole-genome sequencing technologies have now been widely used in bacteriological research [16]. Functional annotation of bacterial genomes can yield a wealth of data at the genetic level [17]. This technology plays an active role in monitoring the genome evolution of pathogenic bacteria and studying the pathogenic mechanisms. Suchawan Pornsukarom et al. used whole-genome technology to perform a comparative analysis of clinical and environmental isolates of *Salmonella* at the phylogenetic level and elucidated the main sources of drug resistance and virulence genes in clinical *Salmonella* isolates [18]. Whole-genome sequencing was also used to monitor the genomic evolution and epidemiological dynamics of the clinical *Campylobacter jejuni* [19]. In this study, we sequenced the whole genome of a *K. pneumoniae* strain whose plasmid carried the colistin resistance gene *MCR-1*. Comparative genomics was used to reveal the epidemic characteristics of IncX4 plasmid carrying *MCR-1*. Functional annotation was performed to elucidate the gene structure of pMJ4-4-MCR plasmid-carrying *MCR-1*. These results will provide new insights into the treatment and control of *K. pneumoniae* infection in hospitals.

## 2. Materials and Methods

### 2.1. Bacterial Isolation and Identification

In August 2021, a strain of *K. pneumoniae*, named KPTA-2108, was isolated from a sputum sample of a patient with lung infections in a tertiary care hospital in Tai’an, China. The sample was cultured on blood agar plate medium (5% sheep blood, LB basal medium) (Autobio, Zhengzhou, China). The strains obtained from the culture were purified and identified by the automated microbial mass spectrometry detection system Autof ms 1000 (Autobio, Zhengzhou, China) according to the procedures of the manual.

### 2.2. Antimicrobial Susceptibility Assay

Antimicrobial susceptibility was tested using two different test methods. The susceptibility to cefazolin, amikacin, cefepime, ciprofloxacin, aztreonam, ceftriaxone, cefuroxime, ertapenem, piperacillin/tazobactam, ampicillin/sulbactam, amoxicillin/clavulanic acid, cefoperazone/sulbactam, gentamicin, cefoxitin, levofloxacin, minocycline, polymyxin, tetracycline, tigecycline, tobramycin, cotrimoxazole, ceftazidime, imipenem, and meropenem was tested with a BD Phoenix M50 fully automated microbial drug sensitivity analyzer (BD, Phoenix, AZ, USA) and MIC values were determined according to the manufacturer’s instruction. The Kirby–Bauer (KB) disc diffusion test determined the sensitivity of cefotaxime and ceftazidime/avibactam. A single colony was resuspended in a normal saline solution, and the turbidity was 0.46~0.54. The bacterial solution was uniformly spread on a Mueller–Hinton agar medium (Autobio, Zhengzhou, China) and a drug-sensitive paper sheet (Thermo Fisher, Waltham, MA, USA) was pasted on the agar plate inoculated with the bacteria to be tested and incubated at 35 °C for 24 h. Results were observed and determined by the size of the antibacterial zone. The drug sensitivity results of the two methods were analyzed according to the 2021 CLSI M100 Executive Standard for Antimicrobial Drug Sensitivity Testing [20,21].

### 2.3. Conjugation Test

The broth mating method was used in the conjugation test. In this method, rifampicin-resistant *E. coli* EC600 was used as the recipient, and the KPTA-2108 strain was used as the donor. The donor and recipient strains were inoculated in 3 mL LB broth (Binhe, Hangzhou, China) and incubated overnight at 37 °C. The donor and recipient bacteria were mixed 1:1 and incubated at 37 °C for 16 to 18 h. The mixture was then applied to LB plates with a concentration of 4 μg/mL colistin and 2.5 mg/mL rifampicin resistance and incubated at 37 °C for 24 h. Finally, single colonies were identified using an automated microbial mass spectrometry detection system Autof ms 1000, and MIC values for colistin were determined using a BD Phoenix M50 fully automated microbial drug sensitivity analyzer.

### 2.4. Bacterial DNA Extraction and Sequencing

PacBio sequencing and analysis were conducted by OE Biotech Co. Ltd. (Shanghai, China). In brief, genomic DNA was extracted from bacteria using the CTAB method [22] according to the manufacturer’s instructions. The genomic DNA was subjected to quality control by agarose gel electrophoresis and quantified by Qubit (Invitrogen, Waltham, MA, USA). The library was constructed with the SMRTbell Template Prep Kit 1.0 (PacBio, Menlo Park, CA, USA). Single-molecule real-time (SMRT) sequencing was performed on the PacBio Sequel II platform using DNA/Polymerase Binding Kit 3.0 (New England Biolabs, Ipswich, MA, USA). Genome assembly was performed using SMRT Analysis 2.3.0 (Pacific Biosciences, Menlo Park, CA, USA).

### 2.5. DNA Sequence Analysis

For the DNA sequence analysis, the genome was assembled by Canu [23,24]. Gene prediction of the assembled genome was performed using Prodigal (v 2.6.3) [25]. The tRNA and rRNA genes were predicted with tRNAscan-SE (v 1.3.1) [26] and RNAmmer (v 1.2) [27]. sRNAs were predicted by BLAST against the Rfam database [28]. The resistance gene of strain KPTA-2108 was annotated in the Comprehensive Antibiotic Resistance Database (CARD) (2020) (https://card.mcmaster.ca/, accessed on 20 April 2022) by running Blast [29]. Multilocus sequence typing (MLST) and capsular serotypes of strain KPTA-2108 were analyzed by Pathogenwatch (v18.3.0) (https://pathogen.watch/, accessed on 19 February 2022) [30]. For typing of plasmids in the KPTA-2108 strain, the PlasmidFinder (v 2.0.1) (https://cge.food.dtu.dk/services/PlasmidFinder/, accessed on 21 April 2022) service platform was used, where plasmid genome sequences were uploaded, and the *Enterobacteriaceae* database was selected with a minimum coverage of 60% [31]. Annotation of virulence genes was done in the Virulence Factors of Pathogenic Bacteria (VFDB) (http://www.mgc.ac.cn/VFs/links.htm, accessed on 20 April 2022) [32]. The annotation of the mobile genetic elements of KPTA-2108 uses the Mobile Element Finder (MEF) (https://cge.food.dtu.dk/services/MobileElementFinder/, accessed on 20 April 2022) [33]. 

### 2.6. Bioinformatics Analyses

In order to analyze the structural features of pMJ4-4-MCR, BLAST Ring Image Generator (BRIG) was used to compare pMJ4-4-MCR with 10 similar plasmids (pS8245-3-CP080094.1, pMCR-WCHEC1606-KY463451.1, pMCR-WCHEC1618-KY463454.1, pMCR-1-Msc-MK172815.1, pDIB-1-MK574665.1, pMCR-NMG38-MK836307.1, pMFDS2258.1-MK869757.1, pMFDS1318.1-MK875282.1, pMFDS1300.1-MK875285.1, and pMCR-1253-A2-MT929278.1) for alignment. A comparative map of the gene structure of pMJ4-4-MCR and the 10 plasmids was produced using Easyfig (v 2.2.5) [34], with the position of *MCR-1* labeled. Three plasmids were randomly selected from these 10 similar plasmids for further analysis. These four plasmids (pMFDS2258.1, pMFDS1318.1, pMFDS1300.1, and pMJ4-4-MCR) were functionally annotated using the NCBI nr library [35,36] and Pfam 35.0 [37] at the protein level, and the bioactive pathways of these plasmids were analyzed using the Kyoto Encyclopedia of Genes and Genomes (KEGG) (https://www.genome.jp/kegg/, accessed on 16 September 2022).

In order to understand the epidemic characteristics of pMJ4-4-MCR, the pMJ4-4-MCR gene sequence was searched in the nt library, and 100 similar complete plasmid sequences were obtained. All similar sequences were aligned by Clustal Omega [38], and a phylogenetic tree was generated using iTOL visualization [39]. The accession numbers and names of these 100 plasmids are shown in Appendix A.

## 3. Results

### 3.1. The Results of the Drug Sensitivity Test

For the strain of *K. pneumoniae* KPTA-2108 isolated in this study, 26 common antibiotics were selected for the drug sensitivity test. As shown in Table 1, the results indicated that the KPTA-2108 strain was resistant to 21 antibiotics, including cephalosporins (cefazolin, cefuroxime, cefepime, cefoxitin, ceftriaxone, cefotaxime, ceftazidime and cefoperazone/sulbactam), fluoroquinolones (ciprofloxacin and levofloxacin), monobactams (aztreonam), penicillins plus β-lactamase inhibitor (amoxicillin/clavulanic acid and ampicillin/sulbactam), aminoglycosides (gentamicin, amikacin and tobramycin), polymyxins (colistin), tetracycline (tetracycline and minocycline), glycylcyclines (tigecycline), and sulfonamides (cotrimoxazole). However, it was sensitive to carbapenems (imipenem, ertapenem, and meropenem), 3rd generation cephalosporins (ceftazidime/avibactam), and antipseudomonal penicillins plusβ-lactamase inhibitors (piperacillin/tazobactam).

The above results indicated that KPTA-2108 was determined to be an MDR strain of *K. pneumoniae* according to the MDR criteria of *Enterobacteriaceae*. In addition, after the conjugation test, single colonies grown on LB plates (4 µg/mL mucin and 2.5 mg/mL rifampicin resistance) were identified as E. coli after strain identification and the drug sensitivity of the recipient bacterium *E. coli* EC600 to colistin was detected, “MIC = 4”. *E. coli* EC600 without conjugation test were sensitive to colistin, “MIC = 1”.

### 3.2. Genome Characterization and Functional Gene Annotation of KPTA-2108

As shown in Table 2, the full length of the KPTA-2108 genome was 5,306,347 bp. The analysis showed that its sequence type (ST) was ST2294 (*gapA*: 4; *infB*: 4; *mdh*: 1; *pgi:* 1; *phoE*: 264; *rpoB*: 4; *tonB*: 10) and capsular Serotypes (K Locus) KL45.

KPTA-2108 had four plasmids, pMJ4-1, pMJ4-2, pMJ4-3, and pMJ4-4-MCR. Of these, pMJ4-1 carried genes for resistance to the main aminoglycoside and sulphonamide antibiotics, and pMJ4-3 carried resistance genes mainly for β-lactam antibiotics. No resistance genes were annotated on pMJ4-2. Of particular note, the colistin-resistance gene *MCR-1* was annotated on the plasmid pMJ4-4-MCR, and it was the only resistance gene on this plasmid that was critical for colistin resistance. Furthermore, conjugation test demonstrated that pMJ4-4-MCR could be horizontally transmitted between bacteria, endowing the recipient with colistin resistance.

Additionally, to further clarify the drug resistance genes carried by this isolate, the chromosomal sequence of KPTA-2108 was annotated in the CARD database for resistance genes. The names of the specific antibiotic resistance genes in the chromosome of strain KPTA-2108 and their registration numbers in the CARD database are as follows: β-lactams [*bla_DHA-1_* (ARO: 3002132), *bla_TEM-1B_* (ARO: 3000014), *bla_SHV-187_* (ARO: 3003154)], quinolones [*qnrB4* (ARO: 3002718), *oqxAB* (ARO: 3003921), *patA* (ARO: 3000024)], aminoglycosides [*rmtB* (ARO: 3000860), *armA* (ARO: 3000858)], macrolides [*mphE* (ARO: 3003741), *msrE* (ARO: 3003109)], polyphosphates [*fosA2* (ARO: 3002804)], tetracyclines [*tetG* (ARO: 3000174)], peptides [*eptA* (ARO: 3003576), *rosA* (ARO: 3003048), *rosB* (ARO: 3003049), *arnA* (ARO: 3002985), *pmrE* (ARO: 3003577)], and sulfonamides [*sul1* (ARO: 3000410)]. However, no carbapenem-resistance genes were found. 

The virulence genes of KPTA-2108 were annotated using the VFDB database. Classifying virulence genes by function, we found virulence genes associated with adherence, antiphagocytosis, efflux pump, iron uptake, nutritional factor, regulation, secretion system, serum resistance, and toxin (Appendix A). In the iron uptake, enterobactin (*entA-F, entS, fepA-D*, *fepG*), salmochelin (*iroE, iroN*), aerobactin (*iutA*), and yersiniabactin (*ybtP)* were found. They have been linked to the pathogenicity of *K. pneumoniae* [40].

### 3.3. Characterization of pMJ4-4-MCR

Plasmid type analysis showed that pMJ4-4-MCR belongs to the IncX4 type, is approximately 30 kb in length and contains only the *MCR-1* resistance gene. Structural alignment with 10 plasmids carrying *MCR-1* revealed that pMJ4-4-MCR had deletions (Figure 1). Although no IS26 family transposase was annotated in pMJ4-4-MCR, successful binding assays showed that the deleted genes did not influence the delivery of colistin-resistance genes in pMJ4-4-MCR.

In addition, pMJ4-4-MCR (accession number: CP107047) aligns with the 10 highest homology plasmids (pS8245-3-CP080094.1,pMCR-WCHEC1606-KY463451.1, pMCR-WCHEC1618-KY463454.1, p*MCR-1*-Msc-MK172815.1, pDIB-1-MK574665.1, pMCR-NMG38-MK836307.1, pMFDS2258.1-MK869757.1, pMFDS1318.1-MK875282.1, pMFDS1300.1-MK875285.1, and pMCR_1253_A2-MT929278.1). Visualization results of comparison were made by BLAST Ring Image Generator (BRIG). The pDIB-1 (33415 bp) was used as the reference sequence, which is the largest plasmid among these 11 plasmids. On the pMJ4-4-MCR coding sequence, the positions of the *pap2* and *MCR-1* resistance genes are also marked. In addition, the location of IS26 was indicated.

The pMJ4-4-MCR (accession number: CP107047) was compared with pMFDS2258.1 (accession number: MK869757.1), pMFDS1318.1 (accession number: MK875282.1) and pMFDS1300.1 (accession number: MK875285.1), followed by a functional annotation. The different IncX4 plasmids carrying *MCR-1* showed significant similarity in the backbone. pMJ4-4-MCR encodes *pap2*, *T4SS*, and *Cag12*. The *pap2* gene was located in the upstream region of *MCR-1* (Figure 2).

The pMJ4-4-MCR (accession number: CP107047) was compared with pMFDS2258.1 (accession number: MK869757.1), pMFDS1318.1 (accession number: MK875282.1), and pMFDS1300.1 (accession number: MK875282.1) for annotation of the genes surrounding *MCR-1* and the type IV secretion system (*T4SS*). Open reading frames are shown as arrows. Green indicates genes associated with mobile elements, red indicates genes associated with resistance, light yellow indicates genes associated with the *T4SS* system, orange indicates other functional genes, hypothetical proteins are marked in gray, and regions with high homology are shaded in pink.

Phylogenetic analysis of the pMJ4-4-MCR plasmid is shown in Figure 3. BLAST comparison revealed 100 similar plasmids to pMJ4-4-MCR. The pMJ4-4-MCR was aligned to 100 IncX4 plasmids carrying *MCR-1*, and a phylogenetic tree was constructed. Plasmids carrying *MCR-1* were found to remain significantly homologous between different strains, suggesting that IncX4-type plasmids can be transmitted horizontally among different strains. The phylogenetic tree showed that the IncX4 plasmid carrying *MCR-1* was more frequent in *E. coli*, and the IncX4 plasmids from *E. coli* strains showed close homology with those from other strains. It is suggested that *E. coli* may be served as the vector of IncX4 plasmid transmission between different bacteria, and the IncX4-type plasmid carrying *MCR-1* in KPTA-2108 may have come from other *Enterobacteriaceae* bacteria. 

The phylogenetic tree was constructed by combining pMJ4-4-MCR with 100 strains of different genera carrying the *MCR-1* plasmids. The different colored fragments represent the different bacterial genera carrying the IncX4 plasmid. The green fragment indicates *E. coli*, the yellow fragment indicates *K. pneumoniae*, the orange fragment indicates *Salmonella* spp., and the blue fragment indicates KPTA-2108.

## 4. Discussion

*K. pneumoniae* is a common hospital-acquired pathogenic microorganism and a serious health problem whose drug resistance has also received wide attention and research. In recent years, isolation of colistin-resistant *K. pneumoniae* has been conducted in different regions. For example, in 2021, Cao et al. reported the coexistence of plasmid-borne colistin resistance gene *MCR-1*, carbapenem resistance gene *bla_NDM-5_*, and *bla_CTX-M-55_* in pneumococcal ST485 isolates, and it can be transmitted in *Enterobacteriaceae* strains [41]. Moreover, some isolates carried multiple *MCR* genes, including *MCR-1*, *MCR-3* and *MCR-8* [42]. In this study, *K. pneumoniae* strain KPTA-2108 carrying the *MCR-1* plasmid was isolated, and whole-genome sequencing of the isolate revealed that the colistin resistance gene *MCR-1* was located in the Incx4-type pMJ4-4MCR plasmid. The presence of the *MCR-1* on the plasmid poses a barrier to its treatment in the clinic. A combination of experimental and previous findings suggested that IncX4-type plasmids carrying *MCR-1* can be transmitted horizontally between strains [43]. Some studies have reported that IncX4 plasmids were highly transmissible, with a transfer frequency 10^2^–10^5^ times higher than that of the prevalent IncFII plasmids [44] and that IncX4 plasmids can be stably maintained in host bacteria, suggesting that IncX4-type plasmid may pose a greater threat than other types [45].

The *MCR-1* gene has been extensively identified worldwide since it was first reported in China in 2015 [46], especially in animals and animal food. There are many large-scale livestock breeding industries in Shandong Province, and colistin has been used for many years as a growth promoter in livestock feed or in the treatment of bacterial infectious diseases such as bovine mastitis [47], leading to a higher isolation rate of pathogenic strains carrying colistin resistance genes in livestock and poultry compared to humans, and most of them were *E. coli* strains [48]. The present study also confirmed that *E. coli* strains in *Enterobacteriaceae* carrying the *MCR-1* resistance gene occurred more frequently than *Salmonella* and *K. pneumoniae*, and they were most likely to play a mediating role in the spread of resistance to other strains. As far as we know, this is the first time that *K. pneumoniae* carrying *MCR-1* plasmid has been isolated from a hospital in the Tai’an area, which suggested that this type of pathogenic bacteria may have already spread in the region, so further epidemiological investigation of *K. pneumoniae* is necessary.

In the case of colistin resistance in KPTA-2108, it was interesting to note that colistin resistance-related genes were annotated both on the plasmid and chromosome. Among them, *eptA*, *rosA*, *rosB*, *arnA* and *pmrE* as endogenous genes on the chromosome have been reported to be related to colistin resistance, their expression can change due to mutations in resistant bacteria [49,50]. But their contribution to colistin resistance was not investigated in depth in this study. According to the results of conjugation and the drug sensitivity tests, the MIC of the recipient *E. coli* EC600 was 4, which was equal to the donor KPTA-2108 (MIC = 4) [51], thus indicating that colistin resistance genes located on the plasmid may have a greater contribution than those on the chromosome. These two different resistance mechanisms were likely to confer synergistic colistin resistance to the *K. pneumoniae* [52]. Genes on the chromosome can affect progeny bacteria through vertical transmission, but genes on the plasmid can spread horizontally between strains of different genera. This raises the prevalence of colistin resistance in *K. pneumoniae* and warns us to pay extra attention to the genetic inheritance and evolution of this resistant bacterium. However, it was noteworthy that a recent article reported that *MCR-1* was found on chromosomes, which indicates that the gene had been integrated and stabilized within the chromosome [53]. 

Recent studies suggested that *K. pneumoniae* had many virulence-related factors, including adherence factors, trophic factors, nutritional factors, secretion systems, and toxins [54]. Among them, increased LPS synthesis and efficiency of iron acquisition through iron carriers were the main factors of enhanced virulence in *K. pneumoniae*. In order to characterize virulence genes carried on the KPTA-2108, the VFDB database was used to obtain functional annotation information of genes related to virulence. Among the plasmids of KPTA-2108, the T4SS secretion system was annotated in the pMJ4-4MCR plasmid, and it could produce sex hairs (F-hairs) and transfer resistance genes between bacteria [55]. It is also the structural basis of plasmid propagation, which explains the success of the conjugation test [56].

We analyzed the relationship between mobile genetic elements on the plasmids and resistance genes (Appendix A). pMJ4-1 carried an abundance of resistance genes, and transposase annotation results showed that the plasmid carried transposases *IS26*, *ISkpn*, *IS903*, and *ISEcl1*. The resistance genes *bla_CTX-M-15_*, *bla_TEM-1B_*, and *qnrS1* in pMJ4-3 were all found to be encoded in the complex transposon ISKpn19. These results illustrate the intrinsic linkage of drug-resistance genes and mobile genetic elements on plasmids [57].

Previous studies have shown that the IncX4 plasmid plays an important role in the propagation of the *MCR-1* gene in Enterobacteriaceae [58]. IncX4 plasmids carrying *MCR-1* showed significant similarity in the backbone [59]. In this study, pMJ4-4MCR was shown to be an IncX4-type plasmid. Functional annotation of the plasmid structure revealed that pMJ4-4MCR encodes the *MCR-1* resistance gene and the *T4SS* secretion system gene and encodes the complete *pap2* gene. Previous reports have shown that *MCR-1*, together with the *pap2* gene, is required for reduced colistin sensitivity [60]. These genes may be responsible for the resistance and horizontal transmission that pMJ4-4MCR has [61]. By aligning pMJ4-4MCR with the most similar plasmids for gene annotation, gene deletions were found in pMJ4-4MCR, but this did not affect the horizontal spread of *MCR-1*.

In conclusion, this study characterized a strain of CPKP isolated for the first time in Tai’an, Shandong Province. With the help of whole-genome sequencing and functional annotation methods, the resistance genes, virulence genes, and mobile genetic elements carried by KPTA-2108 were elucidated. For the IncX4-type pMJ4-4-MCR plasmid, which can be transmitted horizontally, the prevalent characteristics and structural features of pMJ4-4-MCR were analyzed by bioinformatics, and the mechanism of horizontal transfer of pMJ4-4-MCR was attempted to be described. The results showed that KPTA-2108 is an MDR strain with chromosomes carrying an abundance of drug resistance and virulence genes, which is manifested by the multi-drug resistance and pathogenicity to humans of KPTA-2108. IncX4 types carrying *MCR-1* are more frequently prevalent in *E. coli*. pMJ4-4-MCR in the IV secretion system was annotated, and it is closely related to the horizontal transmission of plasmids. The increased of clinically isolated CPKP warns us that the MCR gene has been rapidly spreading before we could prepare for this phenomenon, that more attention should be paid to the prevalence of *MCR-1* in livestock, hospital, and community settings, and that efforts are needed to control the spread of MCR-carrying bacteria.

## Figures and Tables

**Figure 1 pathogens-12-00221-f001:**
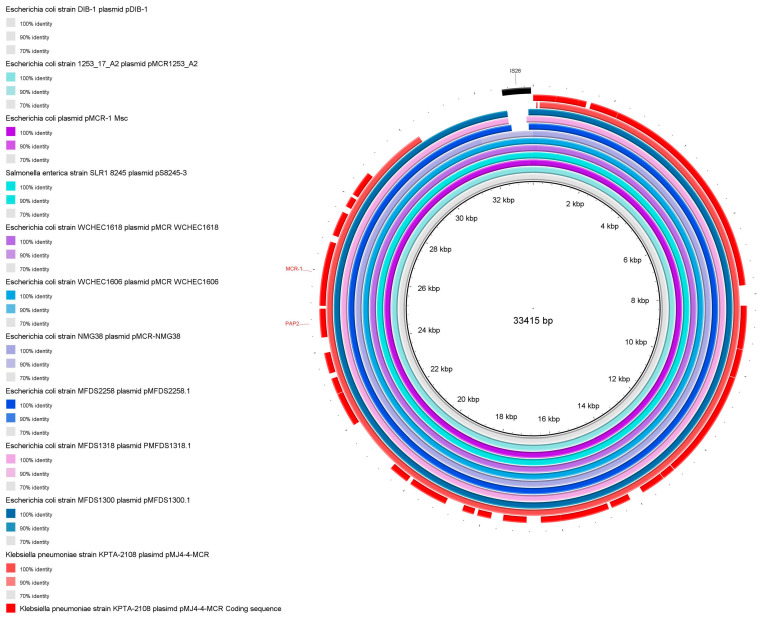
Comparison of the cyclization structure of pMJ4-4-MCR with those of high homology plasmids.

**Figure 2 pathogens-12-00221-f002:**
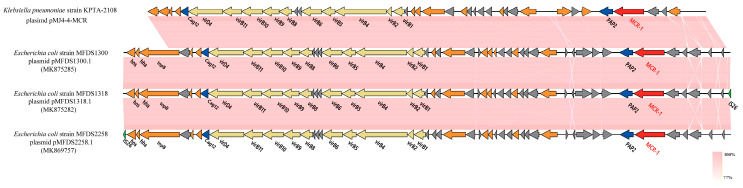
Comparative structural map of pMJ4-4 MCR with other plasmids.

**Figure 3 pathogens-12-00221-f003:**
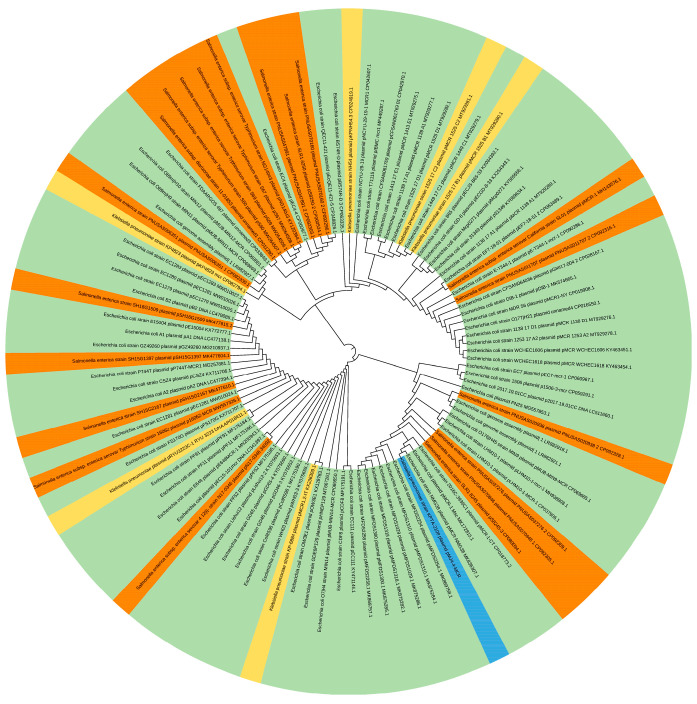
Phylogenetic tree of pMJ4-4 MCR and other lncX4-type plasmids.

**Table 1 pathogens-12-00221-t001:** Drug sensitivity tests of the clinically isolated strain KPTA-2108.

Antibiotics		MIC (μg/mL)	KB (mm)	S/R	Antibiotics		MIC (μg/mL)	KB (mm)	S/R
Cephalosporins	Cefazolin	>16		+	Carbapenems	Ertapenem	0.5		−
	Cefepime	>16		+		Imipenem	≤0.2		−
	Ceftriaxone	>32		+		Meropenem	≤0.1		−
	Cefuroxime	>16		+	Glycylcyclines	Tigecycline	8		+
	Cefoperazone/sulbactam	32/8		+	Sulfonamides	Cotrimoxazole	>4/76		+
	Cefoxitin	>16		+	3rd generation cephalosporins	Ceftazidime/avibactam		25	−
	Ceftazidime	>32		+	Polymyxins	Colistin	4		+
	Cefotaxime		6	+	β-lactamase inhibitors	Piperacillin/tazobactam	16/4		−
Monobactams	Aztreonam	>32		+	Aminoglycosides	Amikacin	>32		+
Fluoroquinolones	Ciprofloxacin	>4		+		Gentamicin	>8		+
	Levofloxacin	>8		+		Tobramycin	>8		+
Penicillins plus β-lactamase inhibitor	Ampicillin/sulbactam	>16/8		+	Tetracyclines	Minocycline	16		+
	Amoxicillin/clavulanic acid	>32/1		+		Tetracycline	>8		+

Note: R indicates resistance (+), S indicates susceptibility (−), KB Kirby–Bauer test.

**Table 2 pathogens-12-00221-t002:** The characterization of genome and plasmids carried by the *K. pneumoniae* KPTA-2108 strain.

Strain/Plasmid	Genome_Size (bp)	Gene_ Num (#)	GC%	rRNA_Num	sRNA_Num	tRNA_Num	DNA Elements	Plasmid Replicon Type	Resistance Genes
Chromosome	5,306,347	4940	0.5741	25	124	88	9	−	−
pMJ4-1	164,378	185	0.5201	0	5	0	0	IncFIB(K)	*bleO*, *aac*, *aadA*, *aph*, *ARR-3*, *mph(A)*, *sul*, *dfrA*, *tet*, *qacE*, *floR*
pMJ4-2	98,026	108	0.4955	0	4	0	2	IncI1-I	−
pMJ4-3	72,739	88	0.524	0	2	0	0	−	*bla_CTX-M_*, *bla_TEM_*, *qnrS1*
pMJ4-4-MCR	30,124	47	0.4131	0	0	0	0	IncX4	*MCR-1*

## Data Availability

The data presented in the study are deposited in the GenBank repository under BioProject number PRJNA887220 and accession number SAMN31156880. The gene sequence accession number of chromosome KPTA-2108 was CP107043, plasimds pMJ4-1: CP107043, pMJ4-2: CP107044:CP107045, pMJ4-3: CP107046, and pMJ4-4MCR: CP107047.

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
