# Peer review of "Genomic Characteristics and Phylogenetic Analyses of a Multiple Drug-Resistant Klebsiella pneumoniae Harboring Plasmid-Mediated MCR-1 Isolated from Tai’an City, China"

_pathogens, 2023, doi:10.3390/pathogens12020221_

Round 1

Reviewer 1 Report

Liu et al. isolated a colistin-resistant Klebsiella pneumoniae strain from a patient in China and analyzed its resistance profile to antibiotics, whole genome sequence and conjugation efficiency. The K. pneumonia strain examined in this study, KPTA-2108, exhibited resistance to 21 antibiotics and harbored four plasmids, including one with the colistin-resistance gene MCR-1, which was transmitted to Escherichia coli. This study will be of great interest to those who study the transmission mechanism of the antibiotic resistance among pathogens, which has gained attraction as a major public health concern. While the findings in this study are quite intriguing, the current manuscript needs extensive revision in various aspects as shown below.

1. Italicize names of bacteria and genes.

2. Pay attention to grammatical errors and typos that hamper readability.

3. Many statements need to be revised to improve clarity. See specific comments.

4. References were not properly added as exemplified by "Error!Reference source not found" that are frequently found in the manuscript, and many statements lack references. Some examples are shown in specific comments but address other instances as well.

5. In the Materials and Methods section, provide manufacturer information and further details for experiments and in silico analyses. See specific comments.

6. For conjugation, the mixture of the donor and recipient cells is usually incubated on the agar plate. However, in this study, it was incubated in the broth medium. Is there any reason for this? Also, although conjugation experiments were mentioned several times as "binding assays" in the Results and Discussion sections, conjugation data were not sufficiently articulated. In fact, I was surprised that no tables or figures were devoted to conjugation data in the manuscript. 

Specific comments:

Title

               Change "extensively-drug resistant" to "extensively drug-resistant".

Abstract

               Lines 16 and 38: Change "Klebsiella (K.) pneumonia" to "Klebsiella pneumoniae".

               Line 17: Change "community and" to "community- and".

               Lines 24-26: Please mention that this resistance profile was obtained via antibiotic resistance assays, not from in silico analysis. Revise "was an extensively drug-resistance (XDR) strain", which seems to be redundant.

               Lines 28-31: Revise the entire statement to improve clarity. Also, the authors need to be cautious in mentioning the role of type IV secretion system since it is not experimentally supported.

Introduction

               Lines 45 and 46: Change "intensive care unit (ICU) acquired" to "intensive care unit (ICU)-acquired".

               Lines 48-50: Provide references.

               Line 61: Start a new paragraph from here.

               Line 62: Remove "abundant data on", which is not necessary.

               Lines 69 and 70: Provide references.

               Line 76: Change "also was used" to "was also used".

               Lines 78 and 79: Change "an isolated K. pneumoniae strain" to "a K. pneumoniae strain".

Materials and Methods

               Line 89: Change "sputum samples" to "a sputum sample".

               Lines 90 and 91: Provide the manufacturer of "Comaga chromogenic medium".

               Line 93: Change "procedure of manual" to "procedures of the manual".

               Lines 102 and 103: Provide further details of the disk diffusion method.

               Lines 103 and 104: Provide references.

               Lines 113 and 114: Clarify "purified by repair”.

               Lines 114-123: These statements are confusing, and some parts appear to be redundant. I strongly suggest re-writing these statements to improve clarity. 

               Lines 126 and 127: This statement is fragmentary. Revise it.

               Line 146: Instead of "some plasmids", describe what plasmids were selected for comparison along with accession nos.

               Lines 146-149: Please clearly mention that 100 similar sequences are entire plasmid sequences, not just MCR-1 gene sequences. 

               Lines 151 and 152: Why were plasmids "pMFDS2258.1, pMFDS1318.1 and pMFDS1300.1" selected for this process? Also, in line 150, "comparative gene structure map" will perplex readers since it is not clear how different it is from "structural analysis of plasmids" in the prior paragraph.

               Lines 153 and 154: Further details are needed for the statement "The four plasmids ... Pfam 35.0". What program was utilized in this process? What does "annotated with functional proteins" exactly mean? How different is this from regular annotation?

               Line 156: Clarify "The results ... structure map".

Results

               Lines 162-165: It will be helpful if the authors group these antibiotics based on their characteristics as shown in lines 185-191 and 267-269.

               Line 179: Change "be critical for colistin-resistance" to "was critical for colistin resistance".

               Line 181: Chang "drug resistance gene" to "drug resistance genes".

               Lines 192 and 193: Change "is due to" to "was due to". Please mention which genes those mutations are located in and provide references for such mutations.

               Lines 197-199: Why was only this group of genes mentioned? How about other groups of virulence genes?

               Lines 204 and 336: I suggest changing "genes deletion" to "deletions".

               Line 237: Revise "Phylogenomic analysis" to "Phylogenetic analysis".

               Line 238: Change "obtained" and "for pMJ4-4-MCR" to "revealed" and "to pMJ4-4-MCR", respectively.

               Lines 242-245: Revise the statement to improve clarity.

               Line 246: Change "may be come" to "may be".

Discussion

               Line 259: Start a new sentence from "for example".

               Line 262: Change "can carry" to "carried".

               Lines 263-266: These statements were already mentioned in the Materials and Methods section; hence, this could be further condensed. In line 263, change "one strain of K. pneumoniae KPTA-2108" to "K. pneumoniae strain KPTA-2108".

               Line 270: Change "annotated the chromosomes" to "annotated the chromosome".

               Line 275: Change "were shown to be" to "were".

               Line 284: Change "compared to a human" to "compared to humans".

               Line 286: more frequently than what? Add "and" before "they were".

               Lines 293 and 294: Clarify "Based on ... in the assay".

               Line 298: What does "cloned bacteria" mean? I suggest replacing it with something else.

               Line 301: Re-write "distribution of this strain" which does not make sense.

               Line 302: Change "indicated" and "had integrated" to "indicates" and "had been integrated".

               Lines 303-305: Provide references. Change "annotated to" to "annotated in".

               Lines 306 and 307: Change "suggested" to "suggests". I recommend removing "had evolved and".

               Line 309: Change "Current studies" to "Recent studies" and "the K. pneumonia" to "K. pneumoniae".

               Lines 310 and 311: Revise "immune ... carriers" since the current list is mixed with virulence mechanisms and virulence factors.

               Lines 312 and 313: Change "K. Pneumonia" to "K. pneumoniae".

               Line 314: Elaborate "further annotation analyses".

               Line 315: Revise "only annotated to a single T4SS" to improve clarity.

               Lines 316 and 317: Was "transmitted ... among bacteria" supported by previous studies or experiments?

               Line 319: Clarify "with the most ... transposase".

               Lines 320-322: I think that this statement ("Overall, ... aggressiveness of the bacteria") is not supported by sufficient data.

               Lines 322 and 323: Provide references for "the virulence genes ... more functional genes".

               Lines 323-327: I am not sure whether these statements are relevant for this study.

               Lines 329 and 330: Why does it matter that pMJ4-4MCR belongs to IncX4-type?

               Lines 331-333: Clarify "Analysis of genes in association with bacterial systemic functions". Overall, this statement seems to be far-fetched since neither reference 47, which reviews type IV secretion systems, nor this study provides data that can validate this statement.

               Lines 337 and 338: Revise "whether ... itself" to improve clarity.

Table 1

               Please explain what "KB" means.

               Condense the table legend and add a symbol to it. Show the symbol where "R" and "S" are located in the table. 

               Please group the antibiotics to facilitate the understanding of readers.

Table 2

               Change "KPTA-2108" to "Chromosome".

Figure 1

               Line 215: Change "was made by" to "were made by".

               Line 216: Change "annotated" to "marked".

Figure 2

               Line 229: Add a space after "MCR".

Figure 3

               Line 253: Add "and" in front of "blue fragment".

Reviewer 2 Report

The authors are highly appreciated for bringing such an important issue forward. The study is interesting; however, as the genome analysis is critical to this study, proper description of the bioinformatic analysis would be great. Thus I am suggesting further  improvement of this section. 

Reviewer 3 Report

Reviewer`s Comments.

The manuscript by Liu et al., titled Genomic characteristics and phylogenetic analyses of a extensively-drug resistant Klebsiella pneumoniae harbouring plasmid-mediated MCR-1 in Tai'an city, China addresses unusual detection plasmid-mediated MCR-1 in Kp. While this is an area that attracts the interest of the scientific community, I sincerely think it is best suited as a short communication as nothing much was discussed at the discussion level. Most of the facts in the results section are replicated or almost replicated in the discussion section thus making it to be a bit repetitive. Find below some of my observations and queries.

Abstract: The spelling; change K. pneumonia to K. pneumoniae.

Introduction:  The phrase error reference was not found, and littered the entire write-up. I guess is a software issue. The authors should re-affirm the statements with cited reference `` However, MCR-1-carrying plasmids have also been reported in many bacteria, such as Salmonella, Escherichia coli, and Acinetobacter baumannii Error! Reference source not found.6. Recently, one study reported that plasmids harbouring MCR-1 can be horizontally transmitted between intestinal flora, allowing recipient bacteria to acquire colistin resistance ability.

The authors should make the objective of the work very clear and concise.

Methodology:  What informed the selection of 26 common antibiotics for the drug sensitivity test?

The manufacturer, country, and place of manufacturing of the medium (Comaga chromogenic 90 medium), for the purification should be indicated.

The authors only provided that the drug sensitivity results were analyzed according to the CLSI M100 Executive Standard for Antimicrobial Drug Sensitivity Testing without year. Besides, the issue of inoculum standardization and the media for sensitivity testing is blank to the reader. Why the authors are trying to conserve words, certain information that will make the work reproducible must be made clear.

Transconjugation Assays: Which method did the authors use, place state the authority for this method. This is too general and vague. Luria-Bertani (LB) broth --- state the manufacturer, country and place of manufacturing.

Bacterial DNA extraction and sequencing: The reference source to be made clearer and reference to the guidelines of the kits used should be spelled out.

Results: The above results suggested that KPTA-2108 is 167 an XDR K. pneumoniae strain what do you mean? Unfortunately, the authors did not clearly define the XDR in the text and in the methodology. So XDR must be defined. E,g MDR is normally defined as the resistance of bacterial isolates to two or more classes of antibiotics ( any member of a  class of antibiotics) etc.  In table 1 the authors indicated MIC?  Was that performed? Or do the authors want readers to infer that it was as described in CLSI M100 Executive Standard for Antimicrobial Drug Sensitivity Testing?  this reviewer considered it not too tidy. The breakpoints etc must be mentioned in the method.

The statement Line 179 ``Furthermore, successful binding assays demonstrated that MCR-1 can be transmitted horizontally between the different strains`` which binding assay is the authors referring to?. Line 191 to 192 The statement ``Interestingly, colistin resistance genes were also annotated in the chromosomes, but their existence due to chromosomal mutations`` what method was used by the authors to ascertain colistin resistance genes are due to mutations?

Line 245: It also suggests that the ncX4-type plasmid carrying MCR-1 in KPTA-2108 may be come from other Enterobacteriaceae bacteria.  Change to ``It also suggests that the IncX4-type plasmid carrying MCR-1 in KPTA-2108 may have come from other Enterobacteriaceae bacteria``

Discussion: The introductory aspect of this discussion is another literature review and methodology adopted in form of a case study which I considered not necessary. It is K. pneumoniae NOT K. pneumonia. Please correct this throughout the discussion and the entire manuscript.

What did the authors trying to say here ``However, whether the genes deletion 337 was due to sequencing or sequence deletion of the bacterium itself was unclear and would 338 be further analyzed in future studies ``

Conclusion: The conclusion is weak and should be concise and contain take-home points with good language.

Round 2

Reviewer 1 Report

I was impressed by the authors' efforts to carefully read and address the suggestions raised by this reviewer and the revised manuscript significantly improved from the previous submission. However, a few suggestions were not sufficiently incorporated as shown below and some revisions were not marked in red in the revised manuscript, which hampered the review process. I also came up with additional, mostly minor, suggestions that need to be handled to further enhance the manuscript. See the specific comments.

Specific comments:

Title: Change "in Tai'an city, China" to "isolated from Tai'an city, China".

Line 75: Change "plasmid-carried" to "plasmid carried".

Line 97: Change "placed" to "resuspended".

Line 97: Replace the comma after "0.46-0.54" with a period. Change "coated in" to "spread on".

Lines 98 and 99: Re-write "coated with a drug-sensitive tablet" for clarity.

Line 115: Re-write "from thalli" for clarity.

Lines 121 and 122: Show the manufacturer of "SMRT Analysis 2.3.0" as well as its geographical location

Line 124: Which assembly software was used? Since this study pertains to only one strain, it does not make sense that Flye or Canu was used to assemble the genome. Also, add a space after "sequence analysis,".

Line 149: Which program was employed for the annotation using the NCBI nr library?

Line 201: Change "in the chromosomes" to "in the chromosome".

Line 214: Change "genes deletion" to "deleted genes".

Line 304: I recommend changing "these strains" to "this bacterium".

Lines 306-308: Add references. This suggestion was also given in the previous review but has not been addressed yet.

Line 312: Change "adherence" and "secretion system" to "adherence factors" and "secretion systems", respectively.

Line 317: Change "the secretion system of T4SS" to "the T4SS secretion system".

Line 317: Replace the comma next to "between bacteria" with a period. Add references.

Lines 323-325: Articulate the main findings from Table S4.

Lines 327 and 328: I believe that it will be helpful to readers if the authors add extra information for IncX4-type plasmid, which they provided in the rebuttal.

Lines 329-331: As I stated in the previous review, this statement is rather far-fetched and needs to be toned down.

Line 334: Change "genes deletion" to "gene deletions".

Table 1: Move the legend in line 171 to below the table. Mark the locations of "R", "S", and "KB" in the table with symbols. This was mentioned in the previous review but has not been addressed sufficiently. Change "colistin" to "Colistin" in the table.
